Effectiveness of antifungal treatments during chytridiomycosis epizootics in populations of an endangered frog

Knapp Roland A. 1 2 roland.knapp@ucsb.edu
http://orcid.org/0000-0002-7745-9990 Joseph Maxwell B. 3
http://orcid.org/0000-0001-7908-438X Smith Thomas C. 1 2
Hegeman Ericka E. 1 2
http://orcid.org/0000-0002-9682-1190 Vredenburg Vance T. 4
Erdman Jr James E. 5
Boiano Daniel M. 6
http://orcid.org/0000-0001-8719-2962 Jani Andrea J. 7
Briggs Cheryl J. 8
1 Sierra Nevada Aquatic Research Laboratory, University of California , Mammoth Lakes, California , United States
2 Earth Research Institute, University of California, Santa Barbara , California , United States
3 Earth Lab, University of Colorado , Boulder, Colorado , United States
4 Department of Biology, San Francisco State University , San Francisco, California , United States
5 California Department of Fish and Wildlife , Bishop, California , United States
6 Sequoia and Kings Canyon National Parks, National Park Service , Three Rivers, California , United States
7 Pacific Biosciences Research Center, University of Hawai’i at Mànoa , Honolulu, Hawai’i , United States
8 Department of Ecology, Evolution, and Marine Biology, University of California , Santa Barbara, California , United States
Bergeron Patrick
Electronic publication date: 2022 Jan 5
Publication date: 2022
Volume: 10
Electronic Location ID: e12712
Received 2021 Jul 29; Accepted 2021 Dec 9
Copyright: © 2022 Knapp et al.
Copyright year: 2022
Copyright holder: Knapp et al.
License: This is an open access article distributed under the terms of the Creative Commons Attribution License, which permits unrestricted use, distribution, reproduction and adaptation in any medium and for any purpose provided that it is properly attributed. For attribution, the original author(s), title, publication source (PeerJ) and either DOI or URL of the article must be cited.
License URL: https://creativecommons.org/licenses/by/4.0/

Keywords: Amphibian chytrid fungus, Batrachochytrium dendrobatidis, Wildlife disease, Epizootic, Host population decline, Antifungal treatment

Funding: Sequoia and Kings Canyon National Parks National Science Foundation (NSF) National Institutes of Health Ecology of Infectious Disease program EF-0723563 NSF Rapid Response Research program IOS-1244804 NSF Long-term Research in Environmental Biology program DEB-1557190 and NSF IOS-1455873 National Park Service P19AC00789 The research described in this paper was supported by grants from Sequoia and Kings Canyon National Parks, National Science Foundation (NSF)–National Institutes of Health Ecology of Infectious Disease program (EF-0723563), NSF Rapid Response Research program (IOS-1244804), NSF Long-term Research in Environmental Biology program (DEB-1557190), and NSF IOS-1455873. Development of this paper was supported by Cooperative Agreement P19AC00789 from the National Park Service. The funders had no role in study design, data collection and analysis, decision to publish, or preparation of the manuscript.

==============================
The recently-emerged amphibian chytrid fungus Batrachochytrium dendrobatidis (Bd) has had an unprecedented impact on global amphibian populations, and highlights the urgent need to develop effective mitigation strategies. We conducted in-situ antifungal treatment experiments in wild populations of the endangered mountain yellow-legged frog during or immediately after Bd-caused mass die-off events. The objective of treatments was to reduce Bd infection intensity (“load”) and in doing so alter frog-Bd dynamics and increase the probability of frog population persistence despite ongoing Bd infection. Experiments included treatment of early life stages (tadpoles and subadults) with the antifungal drug itraconazole, treatment of adults with itraconazole, and augmentation of the skin microbiome of subadults with Janthinobacterium lividum, a commensal bacterium with antifungal properties. All itraconazole treatments caused immediate reductions in Bd load, and produced longer-term effects that differed between life stages. In experiments focused on early life stages, Bd load was reduced in the 2 months immediately following treatment and was associated with increased survival of subadults. However, Bd load and frog survival returned to pre-treatment levels in less than 1 year, and treatment had no effect on population persistence. In adults, treatment reduced Bd load and increased frog survival over the entire 3-year post-treatment period, consistent with frogs having developed an effective adaptive immune response against Bd. Despite this protracted period of reduced impacts of Bd on adults, recruitment into the adult population was limited and the population eventually declined to near-extirpation. In the microbiome augmentation experiment, exposure of subadults to a solution of J. lividum increased concentrations of this potentially protective bacterium on frogs. However, concentrations declined to baseline levels within 1 month and did not have a protective effect against Bd infection. Collectively, these results indicate that our mitigation efforts were ineffective in causing long-term changes in frog-Bd dynamics and increasing population persistence, due largely to the inability of early life stages to mount an effective immune response against Bd. This results in repeated recruitment failure and a low probability of population persistence in the face of ongoing Bd infection.

Introduction

Emerging infectious diseases are increasingly common in wildlife, often due to anthropogenic changes in the ecology of the host or pathogen (Daszak, Cunningham & Hyatt, 2000; Cunningham, Daszak & Wood, 2017). Impacts of disease on wildlife can be severe, including long-term population decline and even extinction, with far-reaching effects on species, communities, and ecosystems (Ostfeld, Keesing & Eviner, 2008; Scheele et al., 2019). Wildlife diseases can also spill over to humans and domestic animals (Alexander et al., 2018). Collectively, these impacts of emerging wildlife diseases have significant consequences to global biodiversity and public health (Daszak, Cunningham & Hyatt, 2000). As such, the ability to manage wildlife diseases is critically important. However, management is often challenging because wildlife diseases are poorly described, few intervention measures are available, and free ranging wildlife are inherently difficult to study and treat (Joseph et al., 2013).

The amphibian disease chytridiomycosis, caused by the chytrid fungus Batrachochytrium dendrobatidis (“Bd”), is one of the most destructive wildlife diseases in recorded history. This recently-emerged fungus (O’Hanlon et al., 2018) is highly pathogenic to a wide range of amphibian taxa and, by one estimate, has caused the severe decline or extinction of at least 500 amphibian species (Scheele et al., 2019), with many more predicted to be at risk (Rödder et al., 2009). The pathogenesis of chytridiomycosis is the result of Bd infection disrupting cutaneous osmoregulatory function, leading to electrolyte imbalance and death (Voyles et al., 2009). Bd infection is transferred between hosts via an aquatic flagellated zoospore stage (Longcore, Pessier & Nichols, 1999).

To reduce the impact of chytridiomycosis and increase the fraction of amphibian hosts that survive chytridiomycosis outbreaks, several mitigation measures have been proposed and tested. These include treating hosts with antifungal agents (i.e., drugs or antifungal symbiotic bacteria), treating the environment with antifungals, and reducing host density (Woodhams et al., 2011; Scheele et al., 2014; Garner et al., 2016). However, existing field trials indicate only short-term benefits to targeted populations and limited effectiveness in promoting host population persistence (Woodhams et al., 2012; Garner et al., 2016). Modeling also suggests that none of these approaches is likely to prevent Bd-driven population extirpation, but reducing Bd loads on individual hosts may have the greatest potential to produce a beneficial outcome (Drawert et al., 2017). Given the uncertainty regarding effectiveness of mitigation measures, there is a critical need for additional field trials.

To evaluate the effectiveness of treatments applied to different host life stages, we took advantage of Bd epizootics occurring in populations of the endangered mountain yellow-legged (“MYL”) frog. MYL frogs are emblematic of global amphibian declines, including those caused by Bd. The MYL frog is a complex of two closely-related species, Rana muscosa and Rana sierrae, endemic to the mountains of California and adjacent Nevada, USA (Vredenburg et al., 2007). During the past century, MYL frogs disappeared from more than 90% of their historical localities (Vredenburg et al., 2007) and are now listed as “endangered” under the U.S. Endangered Species Act (U.S. Fish & Wildlife Service, 2002, 2014). In the Sierra Nevada portion of their range, the primary causes of decline are the introduction of nonnative fish into naturally fishless water bodies and more recently, the spread of Bd (Knapp & Matthews, 2000; Vredenburg et al., 2010). In the absence of Bd, MYL frogs are long-lived, with a 1–3 year tadpole stage and post-metamorphic animals that can live for at least 8–10 years (Matthews & Miaud, 2007). However, they are highly susceptible to chytridiomycosis, and the arrival of Bd in a naive population typically results in rapid increases in Bd prevalence and individual-level infection intensities (“load”), leading to subsequent mass frog mortality (Vredenburg et al., 2010). Such epizootics generally extirpate affected frog populations, and hundreds of such extirpations have occurred in the past several decades as Bd spread across the Sierra Nevada (e.g., Rachowicz et al., 2006; Vredenburg et al., 2010). Examples of affected populations transitioning to an enzootic state, in which host populations and Bd coexist, are rare (Briggs, Knapp & Vredenburg, 2010).

The interaction between host and pathogen influences whether the host population is extirpated by an epizootic or transitions to an enzootic state. In MYL frogs, host population extinction vs persistence can result solely from density-dependent host-pathogen dynamics (Briggs, Knapp & Vredenburg, 2010). This suggests that reducing Bd loads (e.g., by lowering frog density or treating frogs with antifungal agents) could increase both frog survivorship and the likelihood of enzootic dynamics. An adaptive immune response by amphibians against Bd may also facilitate enzootic dynamics (Woodhams et al., 2011). Early life stage amphibians have relatively low immunocompetence (Rollins-Smith, 1998; Grogan et al., 2018b), but adults of some species, including MYL frogs, can develop adaptive immune defenses that may be at least partially protective against Bd (McMahon et al., 2014; Ellison et al., 2015; Grogan et al., 2018a). In theory, antifungal treatments conducted in immunocompetent species and life stages during epizootics could slow the growth of Bd and allow the full development of adaptive immunity, which in turn could increase adult survival and population persistence (Woodhams et al., 2011).

The short duration of antifungal drug application may result in only limited suppression of Bd in hosts and a failure to change the fate of Bd-infected amphibian populations. A more sustained manipulation could potentially produce longer-lasting outcomes. One potential sustained manipulation involves imparting stronger antifungal properties to the skin microbiome of amphibian hosts (Bletz et al., 2013; Woodhams et al., 2014). The feasibility of such a manipulation is suggested by laboratory experiments in which augmentation of the skin microbiome with antifungal bacteria altered frog-Bd dynamics and increased frog survival (Harris et al., 2009; Kueneman et al., 2016; but see also Becker et al., 2011). However, this approach is untested in wild amphibian populations.

Over a decade (2009–2018), we conducted six field trials in an effort to mitigate the impact of epizootics that occurred following Bd spread into naive R. sierrae populations (e.g., Vredenburg et al., 2010). Trials included (1) two treatments of early life stages (tadpoles and recently metamorphosed subadults) with the antifungal drug itraconazole (Garner et al., 2009), (2) three treatments of adults with itraconazole, and (3) one augmentation of the skin microbiome of subadult frogs with Janthinobacterium lividum, a symbiotic bacterium that can occur on amphibian skin and has antifungal properties (Brucker et al., 2008). For all field trials, we predicted that the treatments would reduce Bd load on individuals, increase frog survivorship, and allow the long-term persistence of the treated population despite ongoing Bd infection (i.e., in an enzootic state). We present all six trials together here because the collective results highlight the repeatability of the outcomes, and provide important insights into mechanisms underlying those outcomes. These insights are essential for the development of effective mitigation measures against this devastating wildlife disease.

Methods

The six field trials were conducted at sites located in the remote backcountry of Kings Canyon National Park and Inyo National Forest (California, USA), at elevations of 3,150–3,550 m. These sites are 10 to 26 km from the nearest road, and all supplies were carried in and out on foot. In some cases, this remoteness and wilderness regulations constrained potential study designs. The research was approved by Sequoia and Kings Canyon National Parks (permit number SEKI-2009-SCI-0039, SEKI-2010-SCI-0044, SEKI-2012-SCI-0455, SEKI-2015-SCI-0035), U.S. Fish and Wildlife Service (permit number TE-40090B), U.S. Forest Service (permit number WMD17054), and the Institutional Animal Use and Care Committee at the University of California–Santa Barbara (protocol number 478, 848).

Itraconazole treatment of early life stages

Bd-caused epizootics and resulting mass die-offs of R. sierrae occurred in Barrett Lakes Basin during 2005 to 2007 (Vredenburg et al., 2010) and in Dusy Basin in 2009 (Jani, Knapp & Briggs, 2017). In an effort to prevent the extirpation of remnant populations, itraconazole treatments were conducted during mid-summer of 2009 in Barrett and 2010 in Dusy. Because adults typically succumb to chytridiomycosis early in an epizootic (Vredenburg et al., 2010), at the time of the experiments these populations contained primarily late-stage tadpoles and recently metamorphosed subadults. We used results from basin-wide visual encounter surveys (VES; see Supporting Information: General Methods for details) conducted prior to the experiments to identify the largest remaining populations, and assigned them to treated and control groups at random. The Barrett experiment included three treated and three control populations, and in Dusy, where fewer frog populations remained, a total of three treated and two control populations were used (Table S1). Based on VES conducted before and during the treatments, we estimate that we treated 60–90% of each population.

During the first 3–4 days of the Barrett and Dusy experiments (see below for site-specific details), we captured as many R. sierrae tadpoles and subadults as possible from each pond assigned to the treated group. We held captured animals for the duration of the capture and treatment periods using large mesh pens anchored in the littoral zone of each pond. Following capture, we collected skin swabs from a subset of animals to quantify Bd load (Vredenburg et al., 2010, see Methods–Quantifying Bd load using skin swabs below). In addition, swabbed tadpoles were staged (Gosner, 1960: range = 31–45, median = 41) and subadults were measured (snout-vent length) and weighed. Daily itraconazole treatments were conducted on seven consecutive days (see Supporting Information: General Methods for details). To assess treatment effectiveness in reducing Bd loads, we collected a second set of skin swabs from a subset of animals following the final treatment. After the final treatment, animals were released back into the study ponds. In the control populations, we swabbed tadpoles and subadults on a single day. To ensure that control animals were not swabbed more than once, swabbed animals were held in temporarily-erected pens until swabbing was complete. To quantify the longer-term effects of treatment on Bd load and frog population dynamics, we conducted post-treatment VES and swabbing at each pond, approximately monthly in the treatment year and following year. In treated populations, given that we were unable to capture all animals for treatment, animals swabbed during the post-treatment period likely included a mix of treated and untreated individuals.

The Barrett and Dusy experiments were identical in most respects, but also had some differences. In the Barrett experiment, in the ponds assigned to the treated group, we captured tadpoles and subadults during July 29–August 1, 2009. The 7-day period of daily itraconazole treatments (July 30–August 5) started on the second day of the capture period. Because animals were added to pens throughout the capture period when treatments were underway, animals in the treated group were treated four to seven times. During the treatments, mortality across all three ponds was 6%, and a total of 979 animals completed the treatment and were released back into the study ponds (56, 688, and 235 animals across the three ponds, respectively). For control populations, we collected swabs from one population per day during August 2–4 (total number of swabs collected = 98). We conducted post-treatment VES and swabbing in August and September 2009, and in July, August, and September 2010.

In the Dusy experiment, we captured tadpoles and subadults from the ponds assigned to the treated group during July 24–26, 2010. Daily itraconazole treatments occurred during July 27–August 2, and began after all animals were captured. Therefore, unlike in the Barrett experiment, all animals received seven days of treatment. During the treatments, mortality across all three ponds was 18%, and a total of 3,064 animals completed the treatment and were released back into the study ponds (723, 1,043, and 1,298 animals across the three ponds, respectively). Animals in control ponds were captured, swabbed, and released on July 29 (total number of swabs collected = 80). Follow-up VES in each pond occurred in August and September 2010, and July and August 2011. Unlike in the Barrett experiment, we also collected water samples from all Dusy study ponds before and after the treatments to determine if treatment of frogs reduced the concentration of Bd zoospores in the ponds (“zoospore pool”; Briggs, Knapp & Vredenburg, 2010); all associated methods and results are provided in Supporting Information: Treatment-specific Methods and Results.

Itraconazole treatment of adults

LeConte Basin

In mid-summer 2015, annual disease surveillance at one of the largest remaining Bd-naive R. sierrae populations detected high Bd loads and the presence of many moribund and dead frogs. In response to this epizootic, we conducted two antifungal treatment experiments focused on adults, one in the lower portion of the basin and one in the upper portion (Table S1). The design of the two treatment experiments was nearly identical, differing only in the number of days spent capturing frogs for the “treated” group (Table S2). To simplify logistics, frogs in the treated group were captured during the first 2 to 3 days of the experiment, and frogs for the untreated control group were captured on the following day. Frogs that were visibly sick (as indicated by an impaired righting reflex) were excluded because these frogs were likely within hours of death. In the lower basin, a total of 359 and 102 frogs were captured for the treated and control groups, respectively. In the upper basin, these totals were 206 and 74 frogs. Because of the large size of this population, these totals are likely a relatively small proportion of the total frog population in the basin (maximum VES counts of adults in 2009, 2011, 2012, and 2015 were 4,690, 4,136, 4,074, and 1,047, respectively).

Immediately following capture, frogs were swabbed, tagged with passive integrated transponder (PIT) tags to allow identification of individuals (Joseph & Knapp, 2018), measured, and weighed. Following processing, frogs in the treated category were held in capture date-specific pens for the duration of the 8 to 9 day treatment period and control frogs were released (see Supporting Information: General Methods for details). To determine treatment effectiveness, 93 frogs in the lower basin (31 from each capture date) and 50 frogs in the upper basin (25 from each capture date) were re-swabbed on the penultimate treatment day. After the last treatment, all frogs were released from the pens (285 in the lower basin, 126 in the upper basin, total mortality during treatment = 27%; see Results for additional details).

Post-treatment frog survival was quantified using capture-mark-recapture (CMR) surveys (Joseph & Knapp, 2018) conducted during the summers of 2016, 2017, and 2018. No post-treatment surveys were possible in 2015 because the frog active season was nearly over by the time the treatments were completed. CMR surveys were conducted during 1–3 visits to the study lakes per summer (i.e., primary periods). During each primary period, all frog populations were surveyed on each of 3 consecutive days, except the final period in 2018 when a 1-day survey was conducted. During each daily survey, any adult frogs observed were captured, identified via their PIT tag, and released. When captured for the first time during a primary period, frogs were also swabbed, measured, and weighed. Any untagged frogs captured during the surveys (i.e., frogs that were not part of the initial treatment phase of the experiment; “non-experimental”) were tagged and processed as described above.

Treasure Lakes Basin

In July 2018, we conducted an antifungal treatment of adult R. sierrae in the Treasure Lakes Basin (Table S1). Unlike the LeConte treatments, this treatment was conducted as a management action instead of an experiment due to the advanced stage of the epizootic and the resulting small number of adults remaining in the population. As such, all methods and results are provided in Supporting Information: Treatment-specific Methods and Results.

Microbiome augmentation of subadults

By 2012, the Bd epizootic in Dusy Basin (see above) had caused the extirpation of most R. sierrae populations. Extant populations contained only late-stage tadpoles and recently metamorphosed subadults, and given the absence of any adults, were presumed to represent the final cohorts at these sites. In July 2012, we initiated an experiment focused on subadults at a single pond to test the combined effect of itraconazole treatment and J. lividum augmentation on Bd load and frog survival. This pond was also used in the 2010 experiment in which early life stages were treated with itraconazole (Table S1).

The experiment included a treated group (itraconazole treatment followed by J. lividum exposure) and a control group (no itraconazole, no J. lividum). In designing this experiment, we assumed that any effect of probiotic bacteria would result from protection provided to relatively lightly infected frogs from Bd colonization, and not from reducing the intensity of established heavy infections (R. Harris, 2012, personal communication). Therefore, given that subadults in the study population had high Bd loads, prior to exposing frogs to J. lividum we reduced their Bd loads with a 7-day itraconazole treatment. We did not test independent effects of itraconazole treatment and J. lividum augmentation due to the limited number of subadults available at the study pond.

Subadults were captured on July 12–13 (n = 331), measured, weighed, and assigned at random to treated and control groups at a ratio of approximately 4:1 (271 treated, 60 control). This ratio was chosen to maximize the number of subadults receiving antifungal treatment while maintaining a sufficiently large control group. All animals were given group-specific toe-clips. We used toe clips because PIT tags are too large to be used with subadults, and other tagging methods (e.g., Brannelly, Berger & Skerratt, 2014) tested in pre-experiment trials gave unsatisfactory results. Following processing, subadults in the treated and control groups were held in separate mesh pens. To determine pre-itraconazole Bd loads, a subset of subadults from both groups was swabbed on July 12, and all control animals were released back into the study pond on July 13. Itraconazole treatments were conducted daily during July 12–18 (see Supporting Information: General Methods for details). Of the original 271 subadults, 256 completed the treatment (mortality = 9%) and were available for J. lividum exposure.

To assess the effectiveness of itraconazole treatment in reducing Bd loads and to quantify the amount of J. lividum present naturally on subadults in this population, we swabbed a subset of animals on July 19 immediately prior to J. lividum exposure (see Supporting Information: General Methods for details on the J. lividum qPCR protocol). J. lividum for use in the experiment was obtained from the skin of an adult R. sierrae in Dusy Basin in 2009 and cultured using standard methods (Harris et al., 2009). On July 19, a concentrated solution of J. lividum culture was transported into Dusy Basin on foot in an insulated container. On July 19 and again on July 20, we bathed all itraconazole-treated subadults (n = 256) in a solution of J. lividum culture for 4–4.5 h (75 and 150 mL of J. lividum culture per liter of lake water on July 19 and 20, respectively). No frog mortality occurred during these J. lividum exposures. At the conclusion of the second J. lividum bath, all animals were released back into the pond and the solution of J. lividum culture was carried out of the backcountry and disposed of. Because the survival of J. lividum during the 2 day period of transport and holding is unknown and we were unable to make counts of live cells in the field, the concentration of J. lividum to which frogs were exposed is also unknown.

To evaluate the effects of the combined itraconazole-J. lividum treatment, we surveyed the study population during the summers of 2012 (n = 3 surveys), 2013 (n = 3), 2014 (n = 1), and 2019 (n = 1). During each of these surveys, we conducted VES and captured and swabbed as many subadult and adult frogs as possible, including both experimental (i.e., toe-clipped) and non-experimental (“wild”) animals. The toe-clip, if present, was recorded for each captured individual. Each collected swab was analyzed for both Bd and J. lividum to describe their concentrations on captured frogs.

Quantifying Bd load using skin swabs

We quantified Bd load using standard swabbing and quantitative PCR methods (Boyle et al., 2004; Hyatt et al., 2007). We defined Bd load as the number of ITS1 copies per swab (see Joseph & Knapp, 2018 for details). In post-metamorphic R. sierrae, Bd loads indicative of severe chytridiomycosis are ≥600,000 ITS copies (=5.8 ITS copies on a log10 scale; Vredenburg et al., 2010; Joseph & Knapp, 2018).

Statistical Analyses

We analyzed the results of all treatments using linear simple and multilevel models in a Bayesian framework. All analyses except one used the brms package in R (Bürkner, 2017; Bürkner, 2018; R Core Team, 2020). The exception was the analysis of the CMR data collected as part of the itraconazole treatments in LeConte Basin. The LeConte CMR model was implemented in Stan (Carpenter et al., 2017) directly instead of via the brms interface.

The models described below are the best-fit models that resulted from the workflow outlined in Supporting Information: General Methods. We considered predictors of group- and population-level effects and family-specific parameters to be important when the 95% credible interval (“CI”) of the estimates did not include zero, and relatively unimportant otherwise. We provide the results of all analyses in tabular form, either in the Results for analyses describing the outcome of treatment experiments, or in Supporting Information: Tables for additional analyses. To interpret the coefficients from negative binomial models provided in the tables, note that there is a log link for the mean. In addition, for zero-inflated negative binomial models, there is a logit link for the zero-inflation component. The key results from treatment experiments are also visualized using boxplots or dotplots (when samples sizes were large or relatively small, respectively). When relevant, sample sizes are displayed above the x-axis of each plot. In plots where sample sizes are displayed, the lack of sample size information for a particular group indicates that this group was intentionally not included in surveys and/or sampling. In contrast, a sample size of zero (“n = 0”) indicates that this group was included in surveys and/or sampling, but that no individuals were available for capture and sampling.

Itraconazole treatment of early life stages

For the Barrett and Dusy experiments, we predicted that itraconazole treatment would reduce Bd loads and increase the survival of frogs during and after metamorphosis. In turn, this would result in more subadults counted during VES conducted in treated vs control populations in the year of and the year following treatment. To quantify the effect of treatment on Bd load, we developed separate models to describe (i) pre-treatment differences in Bd loads of the animals assigned to the treated and control groups, (ii) immediate effects of treatment on Bd loads, and (iii) treatment effects on post-release Bd loads in the year of treatment and the following year. Because the treatments in Barrett and Dusy Basins were virtually identical in their design, we combined the results from both experiments into a single dataset, and included basin as a predictor variable in models to account for any between-basin differences.

We evaluated pre-treatment differences in Bd load between treated and control groups using the model bd_load ∼ (treatment × basin) (family = negative binomial, treatment = [treated, control], basin = [Barrett, Dusy]). Life stage (tadpole, subadult) was not included in the model as a predictor because life stage and basin were collinear (i.e., most ponds were dominated by tadpoles but a few contained mostly subadults), and as such we could not estimate their separate effects. Including a group-level effect of site_id did not improve model fit, indicating that between-pond differences were unimportant.

The immediate effect of treatment on Bd load was assessed using the model bd_load ∼ stage + (trt_period × basin) (family = zero-inflated negative binomial, stage = [tadpole, subadult], trt_period = [begin, end of treatment period]). We were able to include life stage in this model because many tadpoles metamorphosed into subadults during the treatment, producing a more balanced representation of life stages across sites. Plots of conditional effects suggested substantial differences in Bd load variation between life stages, treatment categories, and basins. Therefore, the overdispersion parameter was modeled as a function of all three predictor variables.

We evaluated the effect of treatment on post-release Bd loads using the model bd_load ∼ stage + basin + (year_std × treatment) + (1 | site_id) (family = zero-inflated negative binomial, year_std is a dummy variable in which 0 = year of treatment and 1 = year after treatment, site_id included as a group-level effect). Plots of conditional effects suggested substantial differences in Bd load variation between life stages, basins, years, and treatment groups, and therefore the overdispersion parameter was modeled as a function of all four predictor variables.

The effect of treatment on subsequent subadult counts was assessed using the model count ∼ basin + ltadpole + (std_year × treatment) + (1 | site_id) (family = zero-inflated negative binomial, count = number of subadults counted during a post-treatment VES, ltadpole = number of tadpoles counted (log10 transformed) during the same VES, site_id included as a group-level effect). The count of subadults served as a proxy for subadult survival; survival could not be estimated directly using CMR methods because of the inability to tag subadults. We included the tadpole count variable to account for possible differences between ponds in subadult production resulting from differences in the number of tadpoles.

Itraconazole treatment of adults

In the LeConte Basin treatment experiments, to estimate pre-treatment differences in Bd loads of frogs assigned to the treated and control groups, we used the model bd_load ∼ (location × group) (family = negative binomial, location = [lower, upper], group = [treated, control]). To evaluate the immediate effect of treatment on Bd loads, we used the model bd_load ∼ (location × trt_period) (family = negative binomial, trt_period = [begin, end of treatment period]). For both analyses, we excluded any frogs that died during the treatment period. We evaluated differences in Bd loads of frogs that lived vs died during the treatment period using the model trt_died ∼ (lbd_load × location) (family = bernoulli, trt_died = [true, false], lbdload = log10(bd_load + 1) on swabs collected immediately prior to the treatment period).

To describe post-treatment frog population dynamics in the LeConte experiments, including survival and recruitment, while accounting for imperfect detection, we used open population multi-state hidden Markov models (see Supporting Information: Treatment-specific Methods and Results for details). Briefly, we estimated population size over time using parameter-expanded Bayesian data augmentation, which augments the capture histories of observed individuals with a large number of capture histories for individuals that were never detected (Royle & Dorazio, 2012). The states included (1) “not recruited”, (2) “alive at the upper site”, (3) “alive at the lower site”, and (4) “dead”. On any particular survey, we considered three possible observations of an individual: (1) “alive at the upper site”, (2) “alive at the lower site”, and (3) “not detected”. The model structure builds on the work of Joseph & Knapp (2018), tracking individual Bd loads over time, allowing the expected Bd load (log10(Bd load + 1)) to vary as a function of treatment and time, and allowing the effect of Bd load on survival to vary as a function of treatment.

Microbiome augmentation of subadults

To compare pre-treatment Bd loads on frogs assigned to the treated and control groups, we used the model bd_load ∼ expt_trt (family = negative binomial, expt_trt = [treated, control]). The effectiveness of the itraconazole treatment was assessed with the model bd_load ∼ days (family = zero-inflated negative binomial, days = −7 (before treatment) and 0 (after treatment)).

We analyzed the post-treatment data to determine whether subadults exposed to the combined itraconazole-J. lividum treatment had (1) higher concentrations of J. lividum and lower Bd loads than untreated control animals and non-experimental (“wild”) animals, and (2) higher survival than control animals. All analyses focused on data collected during the 2 months immediately following the 2012 treatment. Recaptures of control animals quickly declined to near zero, thereby precluding formal comparisons of J. lividum and Bd load in the treated vs control groups. We were able to compare treated vs wild frogs, but importantly, unlike the treated and control groups that each contained a single cohort of toe-clipped animals that was repeatedly sampled over time, membership of animals in the wild group changed over time as new individuals entered the group following metamorphosis and previously-metamorphosed individuals died. Given this limitation, we describe the J. lividum concentration and Bd load on treated vs control animals graphically only. We analyzed the J. lividum concentration on treated vs wild frogs using the model jliv ∼ (days × frog_group) (family = negative binomial, days = days since J. lividum exposure, frog_group = [treated, wild]). To describe Bd loads of treated vs wild frogs, we used the model bd_load ∼ (days × frog_group) (family = zero-inflated negative binomial). For the second question, we used the percent of animals in each group that were recaptured as a proxy for survival. Formal assessment of the effect of treatment on survival was again not possible due to the rapid disappearance of animals in the control group, so the results are described graphically only.

Results

Itraconazole treatment of early life stages

In the Barrett and Dusy experiments, immediately before itraconazole treatments began, Bd loads of animals in ponds assigned to the treated and control groups were similar (Fig. 1: Week -3 and -1). Model results (Table S3) confirmed that Bd load did not differ between treatment groups. In addition, basin had a weak effect (loads were lower in Dusy than Barrett), and the (treatment × basin) interaction term was unimportant, indicating that the patterns of Bd load between treated and control groups were similar in both basins.

Figure 1 For the itraconazole treatment experiment in Barrett (A) and Dusy (B) basins, temporal patterns of Bd loads of early life stage R. sierrae in populations assigned to control and treated groups.

Weeks -3 and -1 are pre-treatment, week 0 is the end of treatment, and weeks 3–58 are post-treatment. In the boxplots, the horizontal bar is the median, hinges represent first and third quartiles, whiskers extend to the largest and smallest values within 1.5× interquartile range beyond hinges, and dots indicate values outside the 1.5× interquartile range. The number of swabs collected in each week is displayed above the x-axis.

The treatments reduced Bd loads by 1.8 orders of magnitude in Barrett and 6.1 orders of magnitude in Dusy (Fig. 1: Week -1 vs 0). Model results (Table S4) substantiated the important effect of treatment period (“trt_period”; lower at the end than before treatment). In addition, important effects on Bd load were also evident for frog life stage (lower in tadpoles than subadults), basin (higher in Dusy than Barrett), and the (basin × trt_period) interaction term. The importance of the interaction term indicated that loads were higher in Dusy than Barrett at the beginning of the treatment, but lower in Dusy than Barrett at the end of treatment (Fig. 1). Finally, life stage, treatment period, and basin all had important effects on the overdispersion parameter (Table S4; Bd load was more variable in subadults than tadpoles, at the beginning than the end of the treatment period, and in Dusy than Barrett).

After release of the treated animals back into the study ponds, the reduction in Bd load in treated vs control groups that was evident at the end of the treatment period persisted for at least the next 1.5 months (Fig. 1: Week > 0). Results from a model of predictors of Bd load over the 1-year post-release period (Table 1) showed important effects on Bd load of most predictor variables, including treatment (treated lower than control), life stage (lower in tadpoles than subadults), year (lower in the year following treatment (year 1) than the year of treatment (year 0)), and the (year × treatment) interaction term. Basin did not have an important effect. The (year × treatment) term indicated that Bd loads were lower in the treated group than the control group in year 0, but by year 1 loads in the treated group had increased such that Bd loads of the treated and control groups were similar. Therefore, although the treatment effect was evident for more than a month, Bd loads on animals in treated populations returned to pre-treatment levels in the year following treatment (Fig. 1).

Table 1 Effect of itraconazole treatment in Barrett and Dusy basins on Bd loads during the following 1 year period.

Model family is zero-inflated negative binomial.

	Estimate	Est. Error	lo95% CI	up95% CI	Rhat	Bulk ESS	Tail Ess	
Group-level effects	
sd(Intercept)	0.73	0.25	0.40	1.37	1.00	1,301	1,563	
Population-level effects	
Intercept	14.71	0.44	13.87	15.55	1.00	1,676	1,668	
overdispersion-Intercept	−1.33	0.08	−1.48	−1.18	1.00	5,407	2,754	
stage(tadpole)	−2.76	0.10	−2.95	−2.57	1.00	5,296	2,417	
basin(dusy)	−0.26	0.47	−1.17	0.69	1.00	1,966	2,067	
year_std(1)	−0.40	0.12	−0.63	−0.18	1.00	3,922	3,096	
treatment(treated)	−1.32	0.50	−2.33	−0.32	1.00	1,556	1,614	
year_std(1):treatment(treated)	1.52	0.18	1.16	1.87	1.00	3,923	2,645	
overdispersion-stage(tadpole)	0.34	0.07	0.20	0.48	1.00	4,540	3,301	
overdispersion-basin(dusy)	0.45	0.06	0.33	0.57	1.00	5,399	3,119	
overdispersion-year_std(1)	0.82	0.07	0.67	0.96	1.00	4,611	3,192	
overdispersion-treatment(treated)	−0.71	0.07	−0.85	−0.57	1.00	4,632	2,889	
Family-specific parameters	
zi	0.01	0.00	0.01	0.02	1.00	5,437	2,653	

The reduction in Bd load caused by the treatment was associated with increased counts of subadults in treated vs control populations (Fig. 2). Model results (Table 2) indicated that treatment and the (year × treatment) interaction term had important effects. The effects of tadpole count, basin, and year were unimportant. The interaction term indicated that treated populations had higher subadult counts than control populations in the year of the treatment, but that counts in treated populations in the year following treatment were low and similar to those in control populations (Fig. 2). Therefore, mirroring the longer-term effects of treatment on Bd load, the increase in subadult counts in treated populations in the 1.5 months following treatment was no longer evident in the year following treatment.

Figure 2 For control and treated populations in Barrett (A) and Dusy (B) basins, post-treatment counts of R. sierrae subadults in the year the treatment was conducted (year = 0) and the year following the treatment (year = 1).

Each dot indicates the count made during a survey of one of the study ponds, and median values for each treatment group are indicated with a black diamond. The total number of surveys is displayed above the x-axis.

Table 2 Effect of itraconazole treatment in Barrett and Dusy basins on counts of subadults during the following 1 year period.

Model family is negative binomial.

	Estimate	Est. Error	lo95% CI	up95% CI	Rhat	Bulk ESS	Tail Ess	
Group-level effects	
sd(Intercept)	0.33	0.27	0.01	1.00	1.00	1,428	1,686	
Population-level effects	
Intercept	0.76	0.72	−0.57	2.30	1.00	2,810	2,147	
basin(dusy)	0.43	0.50	−0.52	1.43	1.00	3,470	2,893	
ltadpole	0.45	0.24	−0.02	0.91	1.00	3,482	2,531	
year_std(1)	−0.28	0.65	−1.57	0.93	1.00	2,734	2,226	
treatment(treated)	1.65	0.67	0.35	2.97	1.00	2,910	2,770	
year_std(1):treatment(treated)	−2.16	0.86	−3.84	−0.51	1.00	2,546	2,616	
Family-specific parameters	
overdispersion	0.65	0.18	0.37	1.06	1.00	4,460	3,320	

Itraconazole treatment of adults

LeConte Basin

In the two itraconazole treatment experiments conducted in LeConte Basin, prior to the treatment period, adult R. sierrae assigned to the treated and control groups had very high Bd loads (Fig. 3), above the level at which symptoms of severe chytridiomycosis are evident. Bd loads in the control group were somewhat higher than in the treated group, likely because control frogs were captured and processed 1–3 days later than frogs assigned to the treated group (Table S2) and during a period when Bd loads were increasing in the study populations. Model results (Table S6) affirmed an important pre-treatment difference in Bd load between treatment groups (treated groups lower than control groups). Location and the (treatment × location) interaction term were both unimportant, with the latter indicating that the pattern of Bd load between treatment groups was similar in the lower and upper basins.

Figure 3 Effect of itraconazole treatment on Bd loads of adult R. sierrae in the LeConte treatment experiment: (A) lower basin, and (B) upper basin.

The legend for both panels is provided in (B). Box plots show Bd loads on frogs in the control (untreated) and treated groups before the treatment began and at the end of the treatment period. Control frogs were processed and released before the treatment period, and therefore no Bd samples were collected from control frogs at the end of this period. Only frogs that survived to the end of the treatment period and were released back into the study lakes are included. The number of swabs collected from frogs in each category are displayed above the x-axis. Box plot components are as in Fig. 1.

In both experiments, the treatment reduced Bd loads on treated frogs by 1.4–2.7 orders of magnitude (Fig. 3). Model results (Table S7) corroborated the important effect of treatment on Bd load. The effect of location was also important (loads were elevated in upper compared to lower basin), as was the (location × trt_period) interaction term, with the latter indicating that Bd loads before and at the end of treatment were both elevated in the upper compared to the lower basin.

During the treatment period, 74 (21%) of the lower basin treated frogs and 80 (39%) of the upper basin treated frogs died. All control frogs survived during the several hour period between capture, processing, and release. Of the treated frogs that died, most did so during the first half of the treatment period (lower basin: 73%; upper basin: 74%), consistent with frogs succumbing to chytridiomycosis. However, Bd load was not an important predictor of whether frogs died vs survived (Table S8). Location and the (location × bd_load) interaction term were also unimportant.

During the 3-year post-treatment period across both experiments, 213 treated frogs (54%) and two control frogs (1%) were recaptured. We also captured 619 “non-experimental” frogs that were not part of the original treatment experiment. Importantly, the reduced Bd loads that characterized the treated group at the end of the 2015 treatment period (Fig. 3) were maintained in all three post-treatment years (Fig. 4A). Bd load dynamics in control frogs are less clear because of the paucity of control frogs recaptured during 2016–2018. Both recaptured control frogs were recaptured in the year after the treatments (2016), and both had relatively low loads in 2015 relative to the rest of the individuals in the control group (Fig. 4A). During the 2016–2018 period, Bd loads in the non-experimental group were relatively low and similar to those of the treated group. Additional details, including on frog movement patterns and detection probabilities, are provided in Supporting Information: Treatment-specific Methods and Results.

Figure 4 Outcome of the LeConte treatment experiment with adult R. sierrae, showing results for control, treated, and non-experimental animals.

Time series from 2015 to 2018 of (A) observed Bd loads, with lines connecting sequential observations of tagged individuals, (B) posterior estimates for the number of live adults (abundance) in each group, where each point is a draw from the posterior, and (C) estimated relationships between Bd load and adult survival probability during the entire study period, with one line for each posterior draw. A rug along the x-axis displays the observed distributions of Bd load. In (A) and (B), the date tick marks indicate January-01 of each year. In (A) and (C), the Bd load axis shows Bd loads as log10 (copies + 1).

Overall, the LeConte adult population declined in abundance from 2015 to 2018, with the most rapid declines in the control group (Fig. 4B). Between the end of the treatments in 2015 and the first surveys of 2016, the number of animals surviving in the control group dropped from 176 to 8 (CI [2–18]). By the end of the summer in 2016, the posterior median for the number of surviving control animals was 0 (CI [0–0]), indicating an annual survival rate near zero. The rate of decline was slower in both the treated and non-experimental groups (Fig. 4B), and estimated annual survival for 2015–2016, 2016–2017, and 2017–2018 was 0.56, 0.17, and 0.31, respectively. Despite the positive effect of the treatment on survival, by 2018 the study populations in the lower and upper basins had declined to few remaining adults. In the last primary period of 2018, the posterior median for the number of treated frogs alive across both basins was 9 (CI [3–17]), 125 for non-experimental frogs (CI [87–188]), and zero for controls (CI [0–0]).

Bd load had a stronger negative effect on survival in the control group relative to the treated and non-experimental groups (posterior probabilities = 0.99 and 0.99, for control vs treated, and control vs non-experimental, respectively; Fig. 4C). This was the case despite considerable overlap in Bd loads between the control and treated/non-experimental groups (Figs. 4A, 4C). For example, for Bd loads in the range 6–8 (Fig. 4C), control frogs had much lower survival probabilities than did treated and non-experimental frogs.

Treasure Lakes Basin

Results of this non-experimental treatment are provided in Supporting Information: Treatment-specific Methods and Results. Briefly, the itraconazole treatment of adult R. sierrae in the Treasure Lakes Basin reduced Bd loads over the short-term by a similar amount to that observed in the LeConte treatment experiments. Because we were unable to include a control group in this treatment and the number of treated frogs was relatively small, the effect of treatment on subsequent frog survival is uncertain. However, the absence of R. sierrae adults during CMR surveys conducted in the 2 years following the treatment suggests that the reduced Bd loads conferred little or no survival benefit.

Microbiome augmentation of subadults

In the 2012 Dusy Basin microbiome augmentation experiment, subadult frogs assigned to the treated category were treated with itraconazole and then exposed to the J. lividum probiotic. Prior to the itraconazole treatment, Bd loads were similar in subadults assigned to the control and treated groups (Fig. 5: day = −7). Model results (Table S11) affirmed that pre-treatment Bd loads of the two groups were not different. Itraconazole treatment reduced Bd loads almost four orders of magnitude (Fig. 5: day −7 vs 0), and model results (Table S12) substantiated this strong effect.

Figure 5 In the Dusy Basin J. lividum augmentation experiment, temporal patterns of Bd loads on subadult R. sierrae in the control, treated, and wild groups.

Panel labels indicate the number of days since J. lividum exposure. Prior to the exposure of frogs in the treated group to J. lividum on days 0 and 1, frogs in the treated group were treated with itraconazole on days -6 to -1 to reduce their Bd loads. The number of swabs collected on each day is displayed above the x-axis.

Immediately prior to J. lividum exposure, J. lividum concentrations on subadults in the treated group were either zero or near-zero for all individuals (Fig. 6: day 0). A total of 12 days after J. lividum exposure of frogs in the treated group and their release back into the study pond, J. lividum concentrations on subadults in the treated group were high (Fig. 6: day 0 vs 12). Unexpectedly, J. lividum concentrations were also high in the control and wild groups despite their lack of direct J. lividum exposure. Over the following 1.5 months (day 12 to day 56), J. lividum concentrations on subadults declined to near baseline levels (Fig. 6). Formal comparison of J. lividum concentrations from day 12 to day 56 across all three groups was not possible due to the almost complete absence of control frogs on days 37 and 56. However, a model that included the treated and wild groups indicated an important negative effect of the number of days since J. lividum exposure on J. lividum concentration, but no effect of group or the (day × group) interaction term (Table 3). Therefore, J. lividum concentrations declined over the 1.5-month period and at similar rates in both treated and wild frogs.

Figure 6 In the Dusy Basin J. lividum augmentation experiment, temporal patterns of J. lividum concentrations on subadult R. sierrae in the treated, control, and wild groups.

Panel labels indicate the number of days since J. lividum exposure. Prior to the exposure of frogs in the treated group to J. lividum on days 0 and 1, frogs in the treated group were treated with itraconazole on days -6 to -1 to reduce their Bd loads. J. lividum concentrations on day 0 are from samples collected from frogs in the treated group just prior to the first J. lividum exposure. The number of swabs collected on each day is displayed above the x-axis.

Table 3 Effect of number of days since J. lividum exposure and frog group (treated, wild) on J. lividum concentration on frogs.

Model family is negative binomial.

	Estimate	Est. Error	lo95% CI	up95% CI	Rhat	Bulk ESS	Tail Ess	
Population-level effects	
Intercept	7.27	0.33	6.68	7.95	1.00	2,944	2,565	
days	−0.14	0.01	−0.16	−0.12	1.00	2,818	2,525	
frog_group(wild)	−0.29	0.75	−1.71	1.24	1.00	2,222	2,413	
days:frog_group(wild)	0.02	0.02	−0.03	0.06	1.00	2,187	2,082	
Family-specific parameters	
overdispersion	0.33	0.04	0.26	0.41	1.00	3,492	2,373	

Following itraconazole treatment and J. lividum exposure, Bd loads on frogs in the treated group increased despite elevated J. lividum concentrations (as measured on days 12 and 37; Fig. 6), and reached pre-treatment levels after 2 months (day 56; Fig. 5). Increased concentrations of J. lividum measured on control and wild frogs on days 12 and 37 (Fig. 6) also had no obvious effect on Bd load in these groups (Fig. 5). Due to the rapid loss of frogs in the control group, formal comparison of Bd loads from day 12 to day 56 across all three groups was not possible. However, a model that included the treated and wild groups indicated that Bd loads increased during this period, and that Bd loads of wild frogs were higher than those of treated frogs (Table 4). In addition, there was an important effect of the (days × group) interaction term, due to increasing loads of treated frogs vs relatively constant loads of wild frogs. Together, these results indicate that the combined effect of itraconazole treatment and J. lividum exposure was ineffective in preventing the increase in Bd loads to pre-treatment levels.

Table 4 Effect of number of days since J. lividum exposure and frog group (treated, wild) on Bd load on frogs.

Model family is zero-inflated negative binomial.

	Estimate	Est. Error	lo95% CI	up95% CI	Rhat	Bulk ESS	Tail Ess	
Population-level effects	
Intercept	8.42	0.23	7.97	8.89	1.00	3,608	3,126	
days	0.08	0.01	0.07	0.09	1.00	3,854	2,537	
frog_group(wild)	6.74	0.47	5.86	7.68	1.00	2,559	2,131	
days:frog_group(wild)	−0.09	0.01	−0.11	−0.07	1.00	2,567	2,040	
Family-specific parameters	
overdispersion	0.56	0.05	0.47	0.66	1.00	3,773	2,677	
zi	0.08	0.02	0.05	0.12	1.00	4,066	2,525	

During the three surveys of the study population conducted in 2012, we observed a decline in the percent of frogs in the treated and control groups that were recaptured, but the rate of decline was steeper in the control vs treated group (Fig. S4). Although no formal analysis is possible due to the relatively few sample points, the results suggest that the itraconazole-J. lividum treatment increased frog survival over the 2-month period following treatment. Nonetheless, during surveys conducted in 2013 (1 year after treatment), only a single experimental (i.e., toe-clipped) animal was captured. This animal was detected during a survey conducted in early summer, and was a member of the treated group. No R. sierrae of any life stage were detected during surveys in 2014 and 2019. In conclusion, the combined itraconazole treatment and J. lividum exposure neither protected frogs against Bd infection nor increased survival sufficiently to allow persistence of this population over the longer-term.

Discussion

The devastating effect of chytridiomycosis on amphibian populations worldwide (Scheele et al., 2019) highlights the need for effective strategies to mitigate disease impacts in the wild following Bd epizootics. The six field trials we conducted with both early and adult life stages failed to facilitate MYL frog-Bd coexistence. Although all of our antifungal treatments reduced Bd loads on individual frogs for periods ranging from weeks to years, the ultimate outcome of our trials was the decline of the experimental populations to local extirpation or near extirpation following the Bd epizootic. This outcome mirrors that observed in hundreds of wild populations of MYL frogs following Bd epizootics and in which no treatments were conducted (e.g., Rachowicz et al., 2006; Vredenburg et al., 2010), and of field antifungal treatments conducted with other amphibian species (Garner et al., 2016). Nonetheless, compared to alternative strategies of disinfecting habitats or reducing host density (Garner et al., 2016), antifungal treatment of individual hosts is predicted to have the greatest likelihood of a beneficial outcome in reducing the impact of Bd epizootics (Drawert et al., 2017), and is therefore worthy of additional evaluation and refinement.

Itraconazole treatment of early life stages

Our two itraconazole treatment experiments that focused on early life stage R. sierrae (Barrett Lakes and Dusy basins) produced similar short and longer-term outcomes, indicating high repeatability between basins and between years. Despite between-experiment differences in treatment duration, both experiments caused a short-term reduction in Bd loads on individuals and increased frog survival, but loads quickly rebounded and all study populations were eventually extirpated. Similar results were reported by Geiger et al. (2017) following antifungal treatment of tadpoles of the common midwife toad (Alytes obstetricans), and is consistent with tadpoles and subadults having relatively low immunocompetence (Bakar et al., 2016; Grogan et al., 2018b). Hardy et al. (2015) treated recently-metamorphosed Cascades frogs (Rana cascadae) with itraconazole, and reported increased survival of treated animals the following year. This relatively long-term benefit of antifungal treatment was not observed in either of our early life stage treatments of R. sierrae, perhaps indicative of variation in immunocompetence of early life stages between even closely-related species.

Itraconazole treatment of adults

In contrast to the relatively short-lived effects of treating early life stage R. sierrae, our two antifungal treatment experiments in LeConte Basin that focused on R. sierrae adults indicated multi-year effects on Bd load and frog survival. This extended effect appeared to result from both increased disease resistance and tolerance, i.e., the ability to limit pathogen burden and the ability to limit the health impact of a given pathogen burden, respectively (see Schneider & Ayres (2008) and Soares, Teixeira & Moita (2017) for relevant reviews). Increased resistance is suggested by the sustained reduction of Bd loads on treated adults to levels comparable to those on adults in populations that are coexisting with Bd and show enzootic Bd dynamics (Briggs, Knapp & Vredenburg, 2010; Knapp et al., 2011; Joseph & Knapp, 2018). Increased tolerance of Bd infection is inferred by the observation that, for treated and control individuals with similar Bd loads, treated frogs had higher survival than control frogs. This enhanced resistance and tolerance is consistent with treated adults mounting an effective adaptive immune response against Bd (McMahon et al., 2014; Ellison et al., 2015; Grogan et al., 2018a).

Although the antifungal treatment appeared to facilitate an adaptive immune response that reduced Bd loads and increased the survival of treated adults over multiple years, the adult population declined relatively quickly during the post-treatment years, and at the end of the experiment in 2018 few frogs remained. This decline was likely due to both reduced adult survival and low recruitment. Annual survival of treated adults, although higher than that for control frogs, was generally still lower than that of most persisting enzootic MYL frog populations (0.5–0.9: Briggs, Knapp & Vredenburg, 2010; Joseph & Knapp, 2018). The low recruitment of new adults in 2017 and 2018 despite large numbers of early life stage animals in the immediately preceding years resembles recruitment levels we have observed in persistent enzootic R. sierrae populations, and is likely a consequence of high chytridiomycosis-caused mortality of frogs during and soon after metamorphosis (Joseph & Knapp, 2018, and results from Barrett and Dusy treatments described above). Whether the recruitment bottleneck in the LeConte population was more severe than in persistent enzootic MYL frog populations remains an important unanswered question.

The treatment of adults was associated with long-term changes in frog-Bd dynamics, but two results from the experiments complicate the interpretation of the overall treatment effect. First, at the beginning of the experiment, frogs in the control group were captured and processed 1–3 days after frogs in the treated group. Because Bd loads in the population were increasing during this period, Bd loads on control frogs were somewhat higher than those on treated frogs. This could have exaggerated the subsequent differences in survival between control and treated frogs. Although we acknowledge this potential confounding effect, related results suggest that the initial differences in Bd loads between control and treated frogs were not the primary cause of the lower survival of control frogs. Specifically, as mentioned above, for the range of Bd load values that overlapped between frogs in the control and treated groups, treated frogs had much higher survival than control frogs. Therefore, we suggest that the higher survival of treated frogs compared to control frogs during the post-treatment period was primarily due to Bd resistance and tolerance that followed treatment, and not differences in pre-treatment Bd loads between control and treated frogs.

The second complicating result is the unexpectedly large number of non-experimental frogs captured during the post-treatment period. In untreated MYL frog populations, Bd epizootics typically result in the mortality of all, or nearly all, adults within 1 year (Vredenburg et al., 2010). Based on this, if the treatment increased frog survival, as predicted, then during the post-treatment period we would have captured primarily treated frogs, with control frogs and non-experimental frogs being rare or absent. This outcome was observed in the upper basin, where during 2016–2018 we captured 81 treated vs zero control frogs, and only eight non-experimental frogs. Although this same pattern was true in the lower basin for treated and control frogs (132 and two captured, respectively), we also captured 615 non-experimental frogs. These non-experimental frogs could have either survived the 2015 Bd epizootic as adults or recruited into the adult population after the epizootic. Based on their sizes on first capture, most (83%) were older adults that had survived the epizootic, and the remainder were new recruits into the adult population. During the 2016–2018 period, frog-Bd dynamics in this non-experimental group were similar to those of the treated frogs, suggesting that these frogs had also mounted an effective adaptive immune response, and as a result, subsequently showed increased Bd resistance/tolerance and relatively high survival.

The mechanism underlying the unexpectedly high survival of non-experimental frogs in the lower basin during the 2015 epizootic is unknown. In theory, treatment of a large fraction of the adult population could have reduced the pathogen pressure experienced by untreated frogs and increased their survival (Briggs, Knapp & Vredenburg, 2010). However, this would have increased the survival of both non-experimental and control frogs, and in both the lower and upper basins. Instead, only non-experimental frogs were captured in large numbers, and only in the lower basin. Another possible cause of the unexpectedly high survival of non-experimental frogs in the lower basin could be the greater habitat complexity that characterizes the lower basin compared to the upper basin. The upper basin contains a single lake and its associated inlet and outlet streams, and no adjacent ponds, meadows, or springs that provide suitable R. sierrae habitat. As a result, the entire frog population is restricted to the site at which the epizootic occurred. In contrast, the lower basin contains a diverse array of aquatic habitats, including two lakes, four ponds, and associated streams, marshes, and springs, all of which were used by R. sierrae prior to the epizootic. Although conjectural, it is possible that frogs in some of these associated habitats experienced lower pathogen pressure, lower Bd loads, and higher survival during the epizootic than frogs in the much larger lake-dwelling populations. We conclude that although inadequacies in the study design and some paradoxical outcomes complicate the interpretation of the results from the LeConte treatment experiments, the long-term reduction in Bd loads and increased survival of treated frogs are best explained as a direct consequence of the antifungal treatment.

In contrast to the relatively strong effect of the antifungal treatment on adult survival in LeConte Basin, this effect was not observed in the antifungal treatment conducted in the Treasure Lakes Basin. The similar short-term reductions in Bd loads of treated frogs in both the LeConte and Treasure treatments, but the lack of longer term effects on frog survival at Treasure suggests that the strong effects of treatment observed in the LeConte experiments are not universal. Treatment success may depend on the timing of the treatment relative to the onset of the epizootic (later in Treasure than LeConte) or on the inherent susceptibility of the frog population to Bd infection (e.g., Savage & Zamudio, 2011). Because treatment timing is likely important, maximizing the effectiveness of interventions will require frequent disease surveillance of frog populations to detect developing epizootics as early as possible, and rapid adult frog treatments before Bd loads reach high levels. Whether antifungal treatments of adult MYL frogs conducted during this narrow optimal time window would consistently increase long-term frog survival and frog population persistence remains uncertain.

Microbiome augmentation of subadult frogs

Results from the itraconazole treatment experiments (Barrett, Dusy, LeConte) indicate that the effectiveness of these treatments in changing long-term frog-Bd dynamics and facilitating population persistence depends heavily on the survival of subadult frogs and their recruitment into the adult population under post-epizootic conditions. Given the low immunocompetence of subadults against Bd (Rollins-Smith, 1998; Grogan et al., 2018b), short-term itraconazole treatment of early life stages or adults appears insufficient to cause long-term reductions of loads on subadults (i.e., following direct treatment or as a consequence of treatment-altered frog-Bd dynamics). Consequently, subadults succumb to chytridiomycosis relatively quickly and few or none remain to recruit into the adult population. The addition of protective probiotic bacteria to the frog skin microbiome may be a possible means to reduce susceptibility of this vulnerable life stage to chytridiomycosis and increase survival to adulthood (Harris et al., 2009; Bletz et al., 2013; Rebollar, MartÃ­nez-Ugalde & Orta, 2020). In this application, the effectiveness of probiotics will depend critically on the ability of the added bacteria to establish on frog skin and maintain sufficiently high densities over the months or years of the subadult-to-adult transition.

Following our experimental exposure of frogs in Dusy Basin to J. lividum, J. lividum appeared to establish on the skin of frogs in our study population, but subsequently declined to low pre-exposure concentrations and Bd loads increased to lethal levels within two months. Therefore, the predicted protective effect of J. lividum on subadults was not realized. The inability by the probiotic to persist on frog skin may be an important impediment to efforts to augment the microbiome over the long term with species that confer increased protection from Bd (Küng et al., 2014).

An unexpected outcome of the microbiome augmentation experiment was the rapid spread of J. lividum from exposed subadults to control and wild subadults, and its apparent (but short lived) proliferation on the new hosts (J. lividum concentrations on control and wild frogs increased to a level similar to that of frogs that were bathed in J. lividum). Whether J. lividum was transferred via direct frog-to-frog contact or through the water is unknown. Regardless, the spread of J. lividum from exposed to unexposed frogs indicates that if a probiotic with long-term effectiveness against Bd infection is ever identified, its introduction into a frog population may be relatively straight-forward.

Conclusions

The antifungal treatment experiments we describe in this paper were conducted in an effort to increase the likelihood of frog population persistence following Bd epizootics. The results demonstrate that in situ treatment of MYL frogs can strongly reduce Bd loads and increase frog survival over the short term. However, these effects were often emphemeral and failed to facilitate host population persistence in the presence of Bd. Therefore, although reducing Bd loads is an important step, by itself it appears insufficient to facilitate a transition to enzootic host-pathogen dynamics (Briggs, Knapp & Vredenburg, 2010). Effective mitigation of Bd epizootics in MYL frogs will require alternative or combined approaches that cause long-term reduction of Bd loads across multiple life stages and thereby enhance frog survival and recruitment. One possible but untested treatment strategy could be the repeated antifungal treatment of frog populations over several years. Such a “press treatment” would be more effective at keeping Bd loads low for longer periods than our one-time pulse treatments, but would require a substantial (and often unsustainable) commitment of time and resources. In addition, the eventual end of treatment would likely result in the same failure to change long-term frog-Bd dynamics as characterized our pulse treatments.

The lack of effective treatment strategies severely restricts our ability to mitigate the effects of ongoing epizootics in MYL frogs, and poses a serious obstacle to reversing the decline of these and many other threatened amphibian species. One possible solution to this dilemma is additional study of amphibian-Bd systems to better understand what allows population persistence following epizootics (Brannelly et al., 2021). Although preventing the epizootic-caused decline of amphibian populations is a compelling objective, the current lack of any effective and generally applicable strategies to mitigate the impact of Bd epizootics despite the substantial research effort already expended (Garner et al., 2016) serves as an important reminder that preventing population declines by affecting long-term reductions in Bd loads may not be possible with currently-available or reasonably-foreseeable methods. Instead, we suggest that other approaches, including expanding populations that have survived epizootics and are coexisting with Bd in an enzootic state (e.g., Knapp et al., 2016; Joseph & Knapp, 2018; Mendelson, Whitfield & Sredl, 2019) or increasing the capacity of populations to persist despite elevated Bd-caused mortality (Scheele et al., 2014), may produce more immediate conservation benefits for Earth’s many imperiled amphibian species.

Supplemental Information

Supplemental Information 1 Supplemental methods.

Click here for additional data file.

The following people assisted with fieldwork: A. Adams, A. Beechan, D. Burkhart, K. Atkinson, I. Chellman, B. Currinder, C. Dorsey, M. Hernandez, B. Karin, N. Kauffman, A. Killion, J. Lester, A. Lindauer, S. Maple, M. Masten, D. Paolilli, W. Philbrook, G. Ruso, A. Stoerp, and L. Torres. L. Torres developed the J. lividum qPCR protocol while employed in the Vredenburg lab (San Francisco State University). M. Toothman in the Briggs lab (University of California-Santa Barbara) analyzed Bd swabs collected prior to 2016, and K. Rose and A. Barbella in the Knapp lab analyzed swabs collected thereafter.

Additional Information and Declarations

Competing Interests

Author Contributions

Animal Ethics

Field Study Permissions

Data Availability

The authors declare that they have no competing interests.

Roland A. Knapp conceived and designed the experiments, performed the experiments, analyzed the data, prepared figures and/or tables, authored or reviewed drafts of the paper, and approved the final draft.

Maxwell B. Joseph conceived and designed the experiments, performed the experiments, analyzed the data, prepared figures and/or tables, authored or reviewed drafts of the paper, and approved the final draft.

Thomas C. Smith conceived and designed the experiments, performed the experiments, analyzed the data, prepared figures and/or tables, authored or reviewed drafts of the paper, and approved the final draft.

Ericka E. Hegeman performed the experiments, analyzed the data, prepared figures and/or tables, authored or reviewed drafts of the paper, and approved the final draft.

Vance T. Vredenburg conceived and designed the experiments, performed the experiments, authored or reviewed drafts of the paper, and approved the final draft.

James E. Erdman Jr conceived and designed the experiments, performed the experiments, authored or reviewed drafts of the paper, and approved the final draft.

Daniel M. Boiano performed the experiments, authored or reviewed drafts of the paper, and approved the final draft.

Andrea J Jani conceived and designed the experiments, performed the experiments, analyzed the data, prepared figures and/or tables, authored or reviewed drafts of the paper, and approved the final draft.

Cheryl J. Briggs conceived and designed the experiments, authored or reviewed drafts of the paper, and approved the final draft.

The following information was supplied relating to ethical approvals (i.e., approving body and any reference numbers):

Research permits were provided by Sequoia and Kings Canyon National Parks, U.S. Fish and Wildlife Service, U.S. Forest Service, and the Institutional Animal Use and Care Committee at the University of California-Santa Barbara.

The following information was supplied relating to field study approvals (i.e., approving body and any reference numbers):

Permits to conduct disease mitigation research at the study sites were provided by Sequoia and Kings Canyon National Parks and U.S. Forest Service.

The following information was supplied regarding data availability:

All datasets and code to replicate the analyses are available at GitHub: https://github.com/SNARL1/bd-mitigation-report.

All datasets are available at Dryad: Knapp, Roland et al. (2021), Data from: Effectiveness of antifungal treatments during chytridiomycosis epizootics in populations of an endangered frog, Dryad, Dataset, https://doi.org/10.25349/D9D90C.

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
