# Peer review of "Effectiveness of antifungal treatments during chytridiomycosis epizootics in populations of an endangered frog"

_PeerJ, doi:10.7717/peerj.12712_

## Round 0.1 · original submission · Major Revisions

After reading your manuscript, I agree with both reviewers that your paper is well written and provides important information for the conservation of amphibians in the context of Bd.

However, I also agree with reviewer 2 that too much information is presented in the manuscript. Reviewer 1 suggested to add a graphical representation of your different treatments which I think could help. I would suggest pushing this idea further and re-evaluate what information to keep as the take-home of your main results and what to put as supplementary files. For example, the skin microbiome experiment looks like a stand-alone study but not quite at the same time. Asking the reader to constantly go back and forth between the main document and the supplementary files makes the reading more difficult so I think this approach should be harmonized.

Also, we need to know more about the biology of the species (information about natural longevity is fundamental here). It would be important also to clarify the general methods, pit tagging, recapturing and so on before describing the specific experiments. I also found it difficult to keep track of sample sizes and mortality rate (during manipulation and during the experiments) which is, to my point of view quite relevant to disclose, especially in the context of working with an endangered species.

Finally, some of the more descriptive results (e.g. L596-615) could be summarized in a table while results based on very small sample size could presented more concisely.

I see great value in your study and hope that your will find these comments useful.

Reviewer 1 ·

Basic reporting

The paper is well written with a thorough discussion of the results.
Data are clearly presented.

Experimental design

The research question is well defined. The objective of the study was to reduce Bd infection intensity and in doing so alter frog-Bd dynamics and increase the probability of frog population persistence despite ongoing Bd infection. This study yields important insights for future Bd mitigation strategies.

Validity of the findings

This paper describes an interesting study on mitigation of Bd in situ. The objective of the study was to reduce Bd infection intensity and in doing so alter frog-Bd dynamics and increase the probability of frog population persistence despite ongoing Bd infection. This study yields important insights for future Bd mitigation strategies.

Additional comments

Because there are a lot of different treatments I would suggest to add a graphical abstract that presents an overview of the treatments (could be as a supplemental figure).
Miner remarks:
Why is it not possible to determine the bacterial concentration in the J. lividum solution?
Was a necropsy done on the animals that died. Can co-infections, for example with ranavirus, be excluded ?

Reviewer 2 ·

Basic reporting

The level of English was excellent and professional throughout with only minor issues in some areas. The literature references seemed good, though some statements (albeit few) require citations. The use of subtitles helped guide the reader and the figures and tables look professional (although there seems to be minor display issues such as Figure 2 where the “Year” label appears to be cut). I suggest some restructuring of the methods/discussion as some aspects of the experiments became confusing rapidly. I have also suggested reducing the number of figures included in the main manuscript to those that describe the most important results. The manuscript seems to be self-contained and does present results relevant to their hypotheses/predictions but these hypotheses/predictions need to be more clearly stated at the end of the introduction. Further details can be found in my general comments.

Experimental design

This research is undoubtedly necessary in the field and represents original primary research within the aims and scope of the journal. Research questions are important but should be more explicitly stated along with predictions/hypotheses. The knowledge gap based on their literature review is important and this study should help fill a part of it. The investigation did appear to be rigorous and performed to a high technical and ethical standard, although some extra groups of controls would have strengthened the results (see additional comments). Finally, although the methods are described with a fair amount of details, they need to be restructured and simplified with the addition of some minor details (see general comments).

Validity of the findings

This study provides excellent and necessary insight on management techniques relating to frog-Bd dynamics. All the data was provided along with metadata, however due to my lack of knowledge when it comes to Bayesian approaches, I cannot evaluate how sound the presented statistics were. The conclusion is very well stated and followed with a promising management approach, although I would have appreciated a comment relating to pulse vs press treatments.

Additional comments

This manuscript was well written throughout, demonstrating a good level of English. I commend the authors for their extensive research and various approaches they investigated. I do believe this is valuable research that needs to be published. I apologize for the lengthy nature of my comments/questions/advice below, though I hope these will help you revise the manuscript. Please note that the manuscript itself is too long and can/should be considerably shortened. Restructuring the methods and ensuing results/discussion should greatly help in this regard. The addition of link sentences between paragraphs will also greatly improve the flow of the proposed narrative.

L.2-33: The abstract is very well written and structured (but see my comment bellow). I suggest presenting the information in a similar fashion to restructure the methods/discussion sections to improve their clarity.

L.25-29: Your study focuses on two types of treatments performed on different life stages conducted at a fairly restricted time point. Do you think that repeated treatments, treatment of all life stages, and/or the paring of different management efforts (other than your itracanzole + J. lividum trial) may be more beneficial? If you are replicating current strategies or evaluating possible strategies state it clearly and acknowledge that the evaluated strategies did not work but others could. Here you make it sound like treating affected frogs/tadpoles won’t work at all, even though you only covered some strategies. I also think that if space allows, a prompt to investigate other strategies (such as the one described in the conclusion) would close off your abstract well, though this is just a suggestion.

L.34: Good introduction highlighting the importance of studying/managing wildlife diseases.

L.39: Here you write “Diseases of wildlife” which you then switch to “wildlife diseases”. The first formulation is awkward, and I suggest using “wildlife diseases” throughout.

L.42-46: This sentence is a bit long and has some grammatical issues (“many fewer”). I suggest rephrasing it to something like: “As such, the ability to manage wildlife diseases is critically important. However, doing so is often an arduous task considering the lack of information on these diseases, the relatively few intervention measures available, and the inherent difficulty of studying and treating wild animals.”

L.46-47: To say “mostly” would require a citation here considering all the different management strategies for diseases that have come and gone. Otherwise, I suggest rephrasing.

L.48: Add a link to the previous paragraph to improve flow. Something like “One disease that has had worldwide impact is […]”

L.54: I suggest adding a description of the mode of action of this pathogen (how it spreads in the habitat and affect its hosts). It shouldn’t be long, just a few sentences. This is valuable information to the readers and will help them understand points about population density later on. Following this, “In an effort to reduce […]” could be the start of you next paragraph.

L.56: Here you start discussing frogs when just before you were discussing amphibians. I think a link to frogs (maybe something about this group being the most affect, if this is the case) could be made at
L.53-54.

L.59-63: This statement about modeling, though true, does not really fit here. I suggest removing it and moving on to results from laboratory and field studies.

L.63-65: I suggest going a bit further into what these studies found instead of relying on this blanket statement. This sentence also doesn’t really justify your next one since you say that current field tests find limited effectiveness, which does not support your claim of lacking experiments. Perhaps describing the difficulty of evaluating strategies due to the lack of field tests would be a better approach.

L.68: I suggest rephrasing this to improve flow. Something like: "One group of frogs that have been emblematic of amphibian declines caused by Bd are mountain yellow-legged ("MYL") frogs." You need a link with the previous paragraph and following sentence.

L.78: Replace “and” with a comma in “Bd prevalence, infection intensity”

L.79: Remove “epizootic =”

L.86: I think this whole paragraph should be structured differently. The modeling part doesn't add much to point you are making, particularly since you do not develop such models (you are providing experimental evidence). The part about Bd-amphibian dynamics and spread should be higher up when Bd is introduced. If this is done, you could simply refer to the importance of density and load, state it’s backed up by modelling (if needed and without going too much in depth), and briefly describe the potential strategies (L.94-101).

I suggest rephrasing the start of the paragraph with something like this:
"Using empirical data from these rare enzootics and much more common epizootics for modeling revealed that Bd epizootic or enzootic outcomes are governed solely by density-dependent host-pathogen dynamics. This suggests that suppression of Bd loads [...]."

L.90 If kept, “population of frogs”

L.94-99+L.101-111: These parts if combined into a single paragraph highlight very well two strategies which you explore and would be followed up nicely by the third strategy in L.111-119.

L.100-101: The first part of this sentence (up to the comma) is vague and adds nothing to the rest of the sentence. I suggest removing it.

L.199: I would finish this paragraph with a statement on the need for field tests.

L.129: Here you should briefly summarize your predictions/expectations as outlined in your abstract. A brief summary of other field results in the introduction would also help justify these.

L.130-138: I would remove this paragraph altogether as it belongs to the results/discussion and deflates your narrative before it even started.

Methods: I understand that you performed multiple experiments/trials and that concisely summarizing these is challenging, but this section is too long (15 pages long) and must be greatly reduced. Describing general methods as to avoid repetition was a good start, however having these at the start of the methods, without knowing in which context they were performed, lends to confusion. A simple sentence or two at the start of the descriptions summarizing how these methods were applied would help.
You could also start with a general overview of which experiments were performed, how study areas were selected and used (choice of water bodies within basins, etc.) and when they were conducted. This first paragraph will act as anchor point for the rest of the methods and should help guide the reader through these. From there, you could go into general methods, leaving statistical analyses general methods to the end of the methods once all variables have been introduced, along with experiment specific statistic details. Following general methods, you should describe each experiment’s relevant particularities, which would then be followed up by stats and results. Ultimately the format is up to you, but this section needs to be shortened considerably. I have made further suggestions below which may help guide you.

L.144: I would start with a brief sentence of how these surveys are used.

L.144: If you have data on developmental stages of the tadpoles (i.e. Gosner (1960) stage or other; which seems to be the case based on your raw data), I would suggest including these (at least a range of stages). These will be useful in terms of comparisons between studies and to determine how the tadpole's stage may affect its response to treatment.

L.144: Given your definition of an adult vs subadult, this surveying strategy seems arbitrary. Did you measure frogs? Do you have a citation for this way to differentiate developmental stages or is it arbitrary? I believe this requires a citation and a better explanation of how you can assure one frog belonged to one group and not the other.

L.151: Change “During each summer” to “During each experimental summer” or similar wording since you have multiple experiments conducted over multiple years. This statement could be mis-construed as surveying all basins, every year since the first experiment.

L.162-165: I appreciate this statement, but it definitely feels out of place... Maybe include this when appropriate in the results or figure captions? Or perhaps remove the first part of the sentence referring to the figures?

L.170-172: How were subadults and tadpoles ID’ed/ tagged? This would eb a good place to explain this.

L.173-174: At this point I’m wondering which experiments have multi-day treatment periods, which is why a summarizing paragraph at the start might be useful.

L.174-176: What densities of frogs/tadpoles occurred in the pens? We know that tadpoles from certain species of amphibians are affected by population density, with suggestions that this occurs through stress pathways. Could this have affected results? Were pens filled with varying stages/groups or single stage? If multi-stage, could the adults and sub-adults eat the tadpoles opportunistically, which would lower captive/treated tadpole survival? These are questions that could also be avoided if experiments were summarized at the start. I think you should sell the pen for what it is: A way to keep track of wild animals and ensure they receive the proper treatment, reflecting what is done for conservation practices.

L.177: Was this done to prevent resampling of the same individuals? If so, I would state it here as it helps explain the 3-24h range.

L.179-190: Although there is not much you can do about it now, you should have had 2 controls. The first would be your current control, aimed at evaluating how wild frogs react to Bd epizootic when left (mostly) in the wild. The second would have been to verify the effects of the pen (if any) on untreated frogs, to ensure that results aren’t simply dictated by the stress of the pen or other effects it may have. Do you have any citations for the effects of such pens? You should also acknowledge that both forms of controls were feasible and provide an explanation as for why the other wasn’t used (maybe something along the lines of maximizing sample sizes for statistical analyses while also maximizing the number of treated frogs, in light of their imminent Bd related declines). Note that this part feels like it belongs in the discussion too.

L.181-184: The end of this sentence doesn’t make sense given the point you are trying to make.

L.185-186: With the current phrasing this is wrong. If the treatment had no effect, then both groups would be subjected to the same density effects and there would be no treatment bias per se. The bias is in terms of captive controls having lower survival than free controls.

L.189: This is correct, but what about predator-prey dynamics and how they are affected by the pen? Could this cause survival to be higher than controls for instance? Are there predators, and are frogs/tadpoles protected from these while they are in the pens? Is it relevant given the time scale?

L.194-195: I would specify they varied by experiment.

L.197-199: Depending on the experiment or treatment?

L.201: I am not familiar with Bayesian approaches and will thus base my review of this section on comprehension rather than evaluating the statistical approach itself. I would suggest moving all discussions of the statistics to the end of the methods following explanation of the experiments and introduction of variables of interest. This also improves flow from methods to results.

L.211-213: If there is non-independence within/between your treatments (modeled by random effects) it should be accounted for in the full model and can be removed down the line if it doesn't add anything to the model. You need to account for non-independence right away, not "when suggested by the data". This is if sample sizes allow for random effects, which seems to be the case. Of course remember that random effects require at least 5 levels in order to be appropriate and should be fixed effects otherwise.

L.219: There is something wrong with the margins here it seems (in terms of formatting).

L.242-250: This seems like it should be specified in the figure captions rather than the methods.

L.259: Give life stage range if possible.

L.262: Are these populations or sub-populations?

L.265: The prediction should be at the end of your introduction, not in the methods.

L.276: Why the variation? I understand it's the nature of these types of treatments and reflect what is most likely done in the field, but how does this affect results. Are more doses beneficial? Does itraconazole have negative effects if administered too often. How does this variability affect your results? In light of this and other comments, I think your study would benefit from explicitly stating that you are mimicking existing/suggested management strategies (if that is the case), as the design has a large amount of variability that should be controlled for in a classical experiment setup. Then you can easily acknowledge shortcomings as being reflective of current management practices.

L.275: Have you tried using mesh partitions or more pens to separate individuals by day so that they all receive the same treatment?

L.278: What are the samples sizes of these subsets?

L.295: Why was the sampling so unbalanced and how could this affect your results?

L.299: I believe this section may fit better along with the “general methods” statistical paragraph placed at the end of the methods.

L.317: What does this begin-end range mean?

L.317-319: Doesn’t metamorphosis mean that the stages are no longer separate since an individual may have started as a tadpole and ended up in the subadult category? Shouldn’t stage be excluded from your model since the stages aren’t mutually exclusive anymore? I suggest treating stages as a single group of early development frogs, as your sub-title suggests.

L.359: Beyond the difference in sampling effort, do you think sampling a single treatment for the first days and switching over may have biased your results regarding frog capture opportunity through density effects or by selecting healthier, more active frogs early on (particularly if you say that this is a very small proportion of the available population). I'm not sold on the last statement either (do you have population estimates, perhaps from previous studies which could be cited?)

L.365: Why have the general methods if you are going to summarize methods twice in the same paragraph?

L.386: Just say 3 primary periods with the exception of 2018, since it’s the only exception.

L.397: What is “not recruited” in the context of your study?

L.404: Qualitative results are good, but perhaps this could be moved to supplemental materials considering your manuscript is already quite lengthy.

L.453-455: This should be at the end of the introduction as well.

L.475: a subset of how many? Make sure that samples sizes are explicit throughout.

L.481-482: Did you swab controls?

L.491-494: Why alter the concentrations? Also were all subadults bathed twice or separated between days. If separated between days, then this can bring a bias.

L.506: Non-experimental frogs are introduced but have not been mentioned before.

L.525: There are many tables and figures in both the manuscript and supplemental material. In terms of figures, you may want to reduce the number of figures to the important ones regarding your results. In terms of your results, many of them are supported by tables in the supporting documents, and I was wondering if there was a way you could include an abridged version in your actual manuscript for reference, since you do not present model outputs. Note that Figures don’t appear to be displaying correctly (The bottom of “Year” is cut off in Figure 2 for instance).

L.589: The percentages would be appreciated as well.

L.663: I suggest rephrasing since the surprising effect is not relating to the control group dying more, but rather to the non-experimental group performing better.

L.638: Detection probabilities of what?
Discussion: This section also needs to be shortened and I have made some suggestions on how to do so below.

L.713: Your first 1-2 paragraphs should be a recap of why you did this study and a brief summary of your results, which you then discuss in depth in the paragraphs that follow.

L.716-721: I would reword this sentence because it is too lengthy and feels misleading. You could also add a link to your own study and briefly summarize the results you will be discussing afterwards.
"Complete eradication [...], but can be accomplished through disinfection of aquatic habitats and treatment of all amphibians in simple isolated ecosystems. However, similar methods applied in complex environment failed to achieve long-term eradication (citation), highlighting the importance of further investigating management strategies."

L.718: amphibians, and chemical […]

L.722: Move this paragraph to the introduction or remove entirely. The discussion should be used to discuss your results and not bring in new information or justification.
From the previous paragraph rewording suggested, here you could write: "One strategy involves promoting coexistence between amphibians and the pathogen through various strategies such as [...] though these require field testing. In our various experiments we found [...]"

L.736: This information should be in the introduction and not the discussion unless linked with specific results.

L.747: This should probably be part of the first or second paragraph of your discussion. Though it should be re-arranged to improve flow. Start with the theory, then end with the summary of the results.

L.766-769: What about a more sustained/repeated treatment? Here you discuss a pulse event by treating amphibians, but what about pairing this with treatment over multiple years?

L.789: Remove nonetheless

L.796: How long do these frogs typically live in the absence of Bd? Adult survival should decrease every year, so this statement doesn't tell me much unless frogs are supposed to have much higher survival for this time span. I would remove "and by 2018 few treated frogs remained". This question also relates to L.709 where you say no treated frogs remained in 2019 when the experiment was in 2012.

L.801: A range for comparison would be good.

L.803: Here do you have proportions of each life stage encountered? Having 2434 tadpoles, yet very little recruitment is not very surprising for instance, since tadpole survival tends to be low, but for 2434 subadults it would be surprising. I understand that the point is to show frogs are still reproducing but that no new adults seem to be recruited, but because your sampling dates aren't consistent between years, I'd appreciate a more detailed breakdown of your VES by life stage, or at least the VES for subadults.

L.811: In an enzootic, the population is stable, correct? Then knowing we are now in 2021, and the experiment ended in 2018, do you know what happened to the population afterwards? This is to some extent a question you can answer.

L.814: How many of each group?

L.850-852: Good

L.855: Higher than what? Whenever you use this formulation, the thing you are comparing your subject to, should be specified.

L.856-888: Good discussion

L.915-918: So you say that frog microbiota may be resilient to changes, explaining that concentration of J. lividum return to normal. But at the same time, if control frogs can easily acquire a new bacterium over a short period of time, then how resilient is the actual microbiota? It seems like it can easily be disrupted but some mechanisms allow for it to return to normal. Do you have any more in depth explanation for this?

L.962: Reference numbers? Also, although stated in the author notes, just a reminder that this should be placed in the methods.

---

## Round 0.2 · accepted · Accept

Thank you for addressing the comments made by the two previous reviewers and myself.